# Molecular Interaction of Nonsense-Mediated mRNA Decay with Viruses

**DOI:** 10.3390/v15040816

**Published:** 2023-03-23

**Authors:** Md Robel Ahmed, Zhiyou Du

**Affiliations:** College of Life Sciences and Medicine, Zhejiang Sci-Tech University, Hangzhou 310018, China

**Keywords:** nonsense-mediated mRNA decay (NMD), RNA virus, host–virus interaction

## Abstract

The virus–host interaction is dynamic and evolutionary. Viruses have to fight with hosts to establish successful infection. Eukaryotic hosts are equipped with multiple defenses against incoming viruses. One of the host antiviral defenses is the nonsense-mediated mRNA decay (NMD), an evolutionarily conserved mechanism for RNA quality control in eukaryotic cells. NMD ensures the accuracy of mRNA translation by removing the abnormal mRNAs harboring pre-matured stop codons. Many RNA viruses have a genome that contains internal stop codon(s) (iTC). Akin to the premature termination codon in aberrant RNA transcripts, the presence of iTC would activate NMD to degrade iTC-containing viral genomes. A couple of viruses have been reported to be sensitive to the NMD-mediated antiviral defense, while some viruses have evolved with specific *cis*-acting RNA features or *trans*-acting viral proteins to overcome or escape from NMD. Recently, increasing light has been shed on the NMD–virus interaction. This review summarizes the current scenario of NMD-mediated viral RNA degradation and classifies various molecular means by which viruses compromise the NMD-mediated antiviral defense for better infection in their hosts.

## 1. Introduction

Nonsense-mediated mRNA decay (NMD) is an intrinsic mechanism for eukaryotic cells to remove aberrant cellular transcripts harboring a premature termination codon (PTC) [1,2,3] or a long 3′ untranslated region (3′ UTR) [4,5,6,7]. RNA transcripts with PTC created by alternative splicing or mutation produce a non-functional toxic protein that is detrimental to the cells. To maintain stability and functionality, the cellular NMD pathway removes at least part, if not all, of such transcripts and protects the cell from the accumulation of the non-functional mRNA and toxic protein [8,9,10,11]. The degradation and regulation of cellular mRNA by NMD have been identified in various eukaryotes, including *Saccharomyces cerevisiae*, *Drosophila melanogaster*, *Arabidopsis thaliana*, and humans [12]. By targeting non-mutant transcripts, such as mRNAs holding an unusually long 3′ UTR [13], NMD plays a significant role in the growth and development of eukaryotes [14,15,16,17]. Although primarily conceived as a quality control pathway, its regulation of gene expression goes far beyond quality control.

Nascent mRNA transcripts synthesized by RNA polymerase II undergo typical translation, stopping at the correct termination codon near the 3′ end (Figure 1) [18]. Once an unexpected stop codon is introduced in the coding frame, translation terminates ahead, resulting in incomplete translation. Premature termination accelerates the destruction of concerned mRNA through the NMD pathway. The NMD pathway is executed by the sequentially combined action of universally conserved up frameshift protein factors (UPF), eukaryotic release factors (eRF), and suppressor with morphological effect on genitalia (SMG) proteins, in spite of varied functions in different organisms [18]. In the exon junction complex (EJC)-dependent NMD, PTC can trigger the NMD response efficiently when EJC locates at 50–55 nucleotides (nt) upstream of the exon–exon junction [18,19,20,21] (Figure 1). EJC is a dynamic structure installed on pre-mRNAs during splicing and is associated with matured mRNAs until the first round of translation. Usually, the translating ribosome removes EJC from the coding sequences [22,23]. If mRNA translation terminates early due to the presence of PTC, EJC keeps an association with mRNAs, which activates NMD effectors to decay the abnormal mRNA in an EJC-dependent way [24].

Usually, eRF components bind to ribosomes for dynamic translation termination and dissociate the ribosomes from translating templates. In the PTC-containing transcript, EJC is placed 20–24 nt upstream of the exon–exon junction and acts as the anchoring point for the assembly of several NMD components to form the SURF complex (SMG-1-Upf1-eRF1-eRF3) [20], playing a critical role in initiating the NMD pathway [25]. SMG1-mediated phosphorylation of UPF1 triggers the formation of SURF and promotes further steps of NMD activation [26,27], where UPF2 acts as a bridge structure for the connection and assembly of the SURF complex with EJC components [28] (Figure 1). Furthermore, EJC-independent NMD activation is subjected to the 3′ UTR length of a translating mRNA [6,29], which is present in *Drosophila melanogaster, Caenorhabditis elegans, Tetrahymena thermophile*, *Schizosaccarhomyces pombe*, and so on [23,30,31,32]. The long 3′ UTR (>300 nt) delays the ribosome dissociation circumstances and efficient translation termination, such as potential interaction between eRF and PolyA Binding Protein C1 (PABPC1), which leads to the assembly of essential NMD components and corresponding RNA decay [24,33,34]. NMD-mediated degradation of RNA transcripts is initiated by SMG5 and SMG7 in plants or other SMG6-depleted cells [35] or SMG6 in mammals [36]. Subsequently, SMG7 recruits CCR4-NOT by binding with one of its catalytic subunits, which then promotes the deadenylation-dependent decapping [35]. It also recruits decapping complexes such as DCP1, DCP2, and Varicose (VCS), which removes the cap structure of RNA transcripts. The decapped transcripts are further degraded in a 5′→3′ manner through the activity of XRN4 in plants or XRN1 and XRN2 in some other organisms [34,35,37]. A post-transcriptional QC system, NMD is unique and evolutionarily conserved in eukaryotes [38,39,40]. Interestingly, NMD regulates endogenous mRNA turnover but also engages in host defense against virus infection. A broad range of positive-sense RNA viruses are susceptible to and compromised by host NMD.

NMD is a translation-dependent molecular event in eukaryotic cells. Viruses rely on host translation machinery to synthesize viral proteins using viral RNAs as templates. Many RNA viruses have a polycistronic genome that encodes two or more ORFs and is translated to produce all viral proteins via diverse translation strategies. Positive sense (+) genomic RNA can be used as mRNA to translate the first ORF at the 5′ proximity. In some cases, the ribosome can avoid or skip the termination codon of the first ORF and continuously translate to produce a fusion protein via non-canonical translation strategies, such as readthrough and frameshifting [41,42]. Ribosomal frameshifting and readthrough are translation strategies commonly used by many viruses to produce an extended protein, which is usually viral RNA-dependent RNA polymerase [43,44,45]. Once ribosomes keep translation after frameshifting or readthrough, the ribosomal movement would remove the NMD components associated with the coding sequence, thus compromising NMD activation. For polycistronic genomic RNAs, the ORFs near the 3′ end are usually translated via subgenomic RNAs. RNA viruses employing subgenomic RNAs to produce viral proteins have the genomic RNA potentially targeted by NMD because the translation termination of the first 5′ ORF creates an extremely long untranslated sequence. Some viral genomes (e.g., *polioviruses*, HIV, *Dengue virus*, HCV, *potyviruses*) encode a large ORF that is translated to produce a polyprotein. The polyprotein is cleaved by cellular or viral protease enzymes to produce all functional viral proteins [46]. Such viruses use the polyprotein strategy to avoid the presence of any internal stop codon (iTC) in the large ORF, which would protect viral RNAs from NMD. A number of plant viruses or mycoviruses have segmented genomes, some of which are monocistronic. Thus, translation strategies of RNA viruses are closely associated with the activation or escaping of the host NMD defense.

## 2. NMD Defense against Viruses

The host NMD machinery can target genomic and sub-genomic RNAs of RNA viruses if these RNAs exhibit certain NMD-triggering features (Table 1). iTC is a primary feature in many (+) RNA viruses and is responsible for NMD activation [1]. Long 3′ UTRs, retained introns, and multiple ORFs are among the NMD-inducing features in mammalian retrovirus RNAs. These features allow viral RNAs to interact with NMD machinery when translated in host cells (Figure 2) [47]. Some viral RNAs have several NMD-triggering features, leading to a strong NMD response. In addition, NMD-mediated degradation of (+) RNA viruses is independent of the EJC complex since viral RNAs do not undergo RNA splicing [48]. Either upstream ORF or long 3′ UTR plays a vital role in activating EJC-independent NMD [49,50,51]. Other distinctive features accountable for NMD activation might be present in viruses.

### 2.1. NMD Targets Positive Sense RNA Viruses

(+) RNA viruses have their genome translated directly as mRNAs once infect host cells. A majority of RNA viruses have one or more distinctive features, such as polycistronic RNAs, iTC, long 3′ UTR, and uORF, which make (+) RNA viruses susceptible to NMD. For example, *Sindbis Virus* possesses a bicistronic gRNA with iTCs that confront host cellular NMD [52,53]. The unusually long 3′ UTR of *Semliki Forest Virus* (SFV) gRNA was supposed to make the viral gRNA susceptible to NMD, although its RNA has some other features that promote NMD without the long 3′ UTR [54]. *Turnip crinkle virus* (TCV) lacks the 5′ cap structure and a 3′ poly-A tail in its gRNA and uses a non-canonical mechanism to translate its genome. The iTC feature makes the gRNA of TCV sensitive to NMD [55]. iTC and long 3′ UTR in *Potato virus X* (PVX) gRNA and the extended 3′ UTR in sgRNA 1, 2, and 3 accounts for Arabidopsis’s NMD targets [55]. In addition, PVX infection reduces endogenous NMD activity, indicating that PVX may produce anti-NMD protein or make the cellular state hostile to the cellular NMD response [55]. It is a common strategy for RNA viruses to use sub-genomic RNA for expressing viral proteins. Furthermore, compared with host mRNAs, higher GC content in viral 3′ UTR enforces *Tombusviridae* to NMD [56]. *Murine hepatitis virus* (MHV) is a (+) RNA virus belonging to one of the *coronaviruses* that possess NMD-inducing features, such as multiple ORFs with iTC that create a long 3′ UTR during translation [57]. Another case is SARS-CoV2 that is predicted to have a PTC at position 14 of ORF3b in 17.6% of isolates, suggesting that SARS-CoV-2 would be a potential target of NMD [58].

### 2.2. NMD Targets Double-Stranded RNA Viruses

*Rotavirus* is a double-stranded RNA virus displaying interplay with NMD. After entering the host cells, rotaviruses become active and produce 11 kinds of 5′ capped (+) ssRNAs [59]. The key NMD factor UPF1 seems to modulate the expression of viral proteins. The degradation kinetics of rotaviral RNAs is delayed in UPF1 and UPF2 depleted cells and UPF1 and UPF2 are associated with viral RNAs, indicating that viral RNA is sensitive to NMD, although the specific molecular mechanism of interaction between viral RNAs and UPF1 or UPF2 is not identified yet [59].

### 2.3. NMD Targets Retroviruses

Genome organization and lifecycle of Retroviruses differ from those of (+) RNA viruses. Retroviruses occupy multiple ORFs, long 3′ UTR, and retained intron, which can be predicted to trigger NMD (Figure 2b) [47,60]. The termination codon of the *gag* ORF and the iTC of the reverse transcriptase ORF would initiate NMD response for many retroviruses during their translation. *Rous Sarcoma Virus* (RSV) is a retrovirus, and its unspliced gRNA is targeted by NMD due to these NMD-triggering features [61,62,63]. Another retrovirus, *human T-cell leukemia virus type* 1 (HTLV-1) RNA, exposes sensitivity to host NMD [64,65]. Still, the detailed interaction between a specific region of viral RNA and the NMD factor is not well understood. The interaction of the *human immunodeficiency virus* (HIV) with the RNA surveillance pathway is distinctive from other viruses [66]. Here, UPF1 plays an unexpected role in HIV infection [66]. In addition, a stop codon at the termination site of the *gag* gene in *Molony murine leukemia virus* (MoMLV) creates an extremely long 3′ UTR, which makes MoMLV RNA potential for NMD [67].

### 2.4. NMD Targets Viral RNAs of DNA Viruses

The cauliflower mosaic virus (CaMV) belongs to the family *Caulimoviridae* possessing a circular double-stranded DNA genome. The virus replicates via reverse transcription of 35S pre-genomic RNA under the control of the transactivator protein (TAV) [68]. 35S pre-genomic RNA holds six ORFs translated to produce six viral proteins (P1-P6) via the reinitiation mechanism. Alternative splicing is common for the 35S pre-genomic RNA [69], which produces spliced and alternatively spliced mRNAs potentially targeted by NMD [70]. Still, less is known about the mechanism by which NMD targets the CaMV RNA. It can be supposed that multiple uORFs in the 35S pre-genomic RNA may reduce the translation reinitiation efficacy of the original ORFs (P1–P6 coding ORF), which may be responsible for subsequent RNA decay by decay factors [70].

Recently, Zhao and colleagues [71] reported that NMD exerts an antiviral defense against the DNA virus *Kaposi’s sarcoma-associated herpesvirus* (KSHV) by restricting viral lytic reactivation [71]. Using high-throughput transcriptome-wide analysis, they identified viral RNAs as an NMD substrate due to some NMD-eliciting features, such as long 3′ UTRs and introns [71]. These introns facilitate the deposition of EJC in the junction sites and further trigger the EJC-dependent NMD (Figure 2c). Another report suggested that NMD targets the *Rta* transactivator transcripts of KSHV and another herpesvirus, *Epstein-Barr virus* (EBV), for degradation through recognizing features in their 3′ UTRs, which control the lytic reactivation of herpesviruses [72]. Thus, the host NMD system plays an important role in the regulation of the latent-lytic balance during herpesvirus infection.

## 3. Diversity of Viral Tactics Counteracts NMD-Mediated Host Defense

As summarized above, many viruses are targeted by host NMD for degradation. Some viruses have evolved into strategies to counteract the NMD-mediated antiviral defense (Table 1). The viral strategies can be distinct, which depends on viruses. It can be executed via *cis*-acting or *trans*-acting elements. *Cis*-acting elements are mainly viral RNA sequences or structures that impede host NMD. *Trans*-acting molecules are viral proteins via a molecular interaction, which restrain the NMD activating factors.

### 3.1. Viral Proteins: Targeting NMD Components and Interfering with their Function

#### 3.1.1. Movement Proteins

Imamachi and colleagues reported that GC-rich motifs in UPF1 targets were indispensable for UPF1-mediated mRNA decay [56]. Thus, high GC content in the 3′ UTR of viral RNAs could expose viral RNAs to NMD. *Umbravirus Pea enation mosaic virus* 2 (PEMV2) movement protein p26 is required for efficient virus accumulation irrespective of long-distance trafficking. Interestingly, it protects PEMV2 gRNA and NMD-sensitive host mRNAs bearing long, GC-rich 3′ UTRs from NMD [73]. Additionally, p26 safeguards some host cellular mRNAs sensitive to NMD in *N. benthamiana* [73].

#### 3.1.2. Tax and Rex Factors

The Tax factor of the *retrovirus* HTLV-1 attenuates UPF1 and translation commencement factor INT6 activity, both important for NMD [65]. Tax interacts with the central helicase core domain of UPF1 and might plug the RNA channel of UPF1, reducing its binding affinity of nucleic acids [64]. The authors also showed that in the single-molecule approach, Tax’s sequential interaction with an RNA-bound UPF1 freezes and represses UPF1 translocation (Figure 3). In the presence of Tax, the affinity of UPF1 to nucleic acids is reduced by 10 times, and the mutation of UPF1 R843 residue decreases the Tax interaction with UPF1 [64]. The Tax-UPF1 complex accumulates in P-bodies, thus interfering with mRNA degradation [74]. The SMG5/SMG7 complex recycles UPF1 by dephosphorylating via PP2A in P bodies [75]. The Tax-UPF1 complex located in P-bodies indicates that Tax inhibits UPF1 recycling [74]. Tax interacts with other NMD factors, including UPF2 and INT6/eIF3E, preventing the degradation of some cellular and viral RNAs by NMD [65,76]. HTLV-1 Rex protein involved in the nuclear transport of spliced and unspliced viral protein also has an NMD inhibiting role and protects host and viral RNAs from being degraded [21,77]. Rex binds with multiple NMD proteins such as UPF1, SMG5, and SMG7 and selectively replaces UPF3B with UPF3A (a less active form of UPF3) to inhibit NMD [21].

#### 3.1.3. Capsid Protein (CP)

*West Nile Virus* (WNV) belongs to the genus *flavivirus*, closely related to *Dengue virus* and *Zika virus*. The WNV genome encodes some proteins that participate in the down-regulation of host NMD. RBM8A is an EJC protein directly associated with WNV RNA and induces its decay, while PYM1 is an EJC recycling factor that binds with RBM8A to sharpen the decay circumstances. The knockdown of PYM1 attenuates RBM8A binding with WNV RNA, indicating that the PYM1-RBM8A interaction plays a role in NMD against WNV [78]. Hence, the WNV CP protein interacts with PYM1 and restrains it from association with RBM8A, circumventing NMD [78]. RNAi screening-based study also demonstrates that the components (e.g., PYM1) associated with EJC-recycling and NMD pathway are functional in defense against WNV, *Dengue virus*, and *Zika virus*. *Zika virus* is a single-stranded positive-sense RNA virus responsible for severe neurological complications. It infects host cells by impairing endogenous NMD turnover. The CP protein of *Zika virus* depletes the level of UPF1, ultimately inactivating NMD [79]. By association with UPF1, *Zika virus* CP induces proteasome-mediated UPF1 degradation [80]. Recently, evidence has been found that SFV CP also suppresses NMD in translation-independent manner [81]. SFV CP binds with UPF1 and other viral proteins, but the detailed mechanism is not yet elucidated [81].

#### 3.1.4. Core Protein

Core protein from *Hepatitis C Virus* (HCV) also down-regulates endogenous NMD turnover pathway. It disrupts the interaction between WIBG and the EJC component MAGOH/Y14, thus overwhelming NMD [82]. WIBG is a host protein participating in the cytoplasm’s EJC disassembly and recycling process. Generally, WIBG binds MAGOH/Y14 to recycle EJC and accelerates the recycled EJC back to nuclei. By interacting with WIBG to interrupt EJC recycling, the HCV core protein perturbs EJC back to the nucleus from the cytoplasm (Figure 3) [82]. The WIBG–HCV core protein interaction possibly alters cellular transcriptome favoring HCV infection, which requires further study.

#### 3.1.5. Trans-Activator Protein (TAV)

CaMV TAV acts as a suppressor of NMD by stimulating the reinitiation of viral RNA translation. As aforementioned (Section 3.2), failure to reinitiate translation of 35S pre-genomic RNA triggers RNA decay. TAV helps to accomplish translation reinitiation of 35S polycistronic mRNA of CaMV by recruiting the reinitiation supporting protein (RISP), eukaryotic translation initiation factor (eIF3), and other translation components at the translation site. To maintain the highest catalytic activity of RISP, eIF3, and other proteins involved in reinitiation, TAV activates rapamycin (TOR) targets [83,84,85,86], which represses the translation reinitiation barrier created by multiple uORFs in the 35S pre-genomic RNA [87]. In addition, TAV potentially interacts with the decapping complex scaffold protein VCS and protects NMD target transcripts containing uORFs within their leader regions (Figure 3) [70]. VCS contributes to more than 50% of short-live mRNA decay in *Arabidopsis* [88].

#### 3.1.6. N-Protein

Evasion of NMD response is mandatory for Coronavirus (CoV) to establish successful infection. Several proteins of *Coronavirus* have been characterized to be involved in viral RNA package [89,90,91,92] and also help CoV to interact with host silencing factors for efficient replication. Coronavirus N protein plays multifunctional roles in the virus lifecycle. The MHV N protein represses the host NMD pathway and helps the virus to overcome the host defense [57]. In silico analysis suggests that the N protein of the chicken coronavirus *infectious bronchitis virus* (IBV) possibly binds to the NMD core factor UPF1 and compromises its activity, thus antagonizing NMD (Figure 3) [93]. Recently, protein interactome revealed that SARS-CoV-2 N protein interacts with mRNA decay factors UPF1 and MOV10, which might be one of the strategies for SARS-CoV-2 compromising NMD [58].

#### 3.1.7. Replicase Protein

The genome of (+) RNA viruses is the template for translation to produce viral proteins, but it also is recruited into viral replicating compartments as the template for viral RNA replication. Thus, there must be a competition for viral gRNA between viral RNA translation and replication. NMD is dependent on RNA translation. Balistreri and colleagues found that the knockdown of NMD proteins (UPF1, Smg5, Smg7) markedly increased the concentration of SFV RNA and proteins, and the mutation or deletion of viral replicase enhanced NMD response [54]. Accordingly, a competition model was proposed for counteracting NMD-mediated antiviral defense, where SFV replicase protein competes with cellular UPF1 to bind (+) viral RNA by dislocating UPF1 from viral RNA inside the host cell (Figure 3) [54,80].

#### 3.1.8. Non-Structural Protein

Recent evidence reported that non-structural protein from *rotavirus* circumvented the NMD pathway in a strain-independent manner [59]. Viral non-structural protein 5 (NSP5) binds with cellular UPF1 and activates proteasome-mediated degradation of UPF1 (Figure 3) [59].

### 3.2. Viral RNA Sequences or Structures

Besides *trans*-acting viral proteins, viruses have evolved to counteract the NMD-mediated antiviral defense via *cis*-acting RNA elements. These RNA elements include some RNA structures involved in viral non-canonical RNA translation, such as leaky scanning, readthrough, frameshifting and reinitiating, and some specialized RNA sequences engaged in RNA: protein interaction to restrict UPF1 binding [1,94]. However, some *cis*-acting RNA elements directly interact with core NMD components to overwhelm RNA decay [47,94,95].

#### 3.2.1. *Rous Sarcoma Virus* RNA Stability Element (RSE)

Unspliced full-length RSV RNA is entirely resistant to NMD [63], but the insertion of PTC in the RSV molecular clone makes it an NMD substrate [61]. Several follow-up studies show that RSV evades NMD and expresses essential gag protein for virion synthesis [96]. A specific *cis*-acting element called RNA stability element (RSE) protects the RSV gRNA from host NMD. RSE restricts UPF1 to bind eRF3 by recruiting polypyrimidine tract binding protein 1 (PTBP1) [47,62]. Mutations or lack of RSE impose viral genome decay. This study also suggests that the 3′ UTRs of RSV *pol* and *src* may also function as stability elements. Increasing the stability of viral RNAs in cytoplasm also helps *vaccinia virus* and other poxviruses to overcome RNA decay [97]. Similarly, PTBP1 shields retroviral and cellular RNAs from NMD degradation [98]. PTBP1 restricts UPF1 binding and regulates gene expression by locating at the stop codon’s vicinity.

#### 3.2.2. Unstructured Element of TCV

TCV has a (+) single-stranded gRNA, which can be used as a template to express p33 and its readthrough product p94, which are viral replicases. TCV transcribes to create two sgRNAs that are translated to produce viral CP and MP proteins. Both gRNA or sgRNAs of TCV are resistant to NMD in *N. benthamiana* [1]. A 51 nt long unstructured region (USR) at the beginning of the TCV 3′ UTR was determined to be an RSE for sgRNAs and unrelated NMD-sensitive RNAs. Resistance was abolished while introducing two bases in the element to create a stable hairpin. Another RSE is the readthrough element located downstream of the TCV p28 termination codon, stabilizing an NMD-sensitive reporter transcript from decay [1]. This study also demonstrates that viral sgRNA1 lacking 5′cap is NMD resistant, while the addition of 5′cap makes the RNA sensitive to NMD.

#### 3.2.3. Readthrough and Frameshifting Elements

Readthrough elements from *Moloney murine leukemia virus* (MoMLV) and *Colorado tick fever virus* (CTFV) protect sensitive transcripts from NMD [67,99]. The preliminary stage of NMD is initiated by eRF1 and eRF3, which then recruits UPF1. MoMLV reverse transcriptase (MoMLV-RT), interacts with the C terminal domain of eRF1 via its RNase H domain and blocks it from binding with eRF3. The RT-eRF1 interaction promotes mRNA’s readthrough and escapes from NMD [67]. Inserting readthrough stimulatory element and MoMLV PseudoKnot (MLVPK) into an NMD reporter transcript protects it from NMD [5]. Further study reveals that MLVPK antagonizes UPF1 recruitment by eRF1 and subsequent NMD pathway. Another readthrough element is a hairpin structure from CTFV which allows ribosomes to pass the termination codon for continued scanning, thus preventing NMD [99,100].

Ribosomal frameshifting is the translation process where translating ribosomes move forward or step back by one or two nucleotides to escape in-frame stop codon, allowing ribosomes to translate continuously to produce a large fusion protein. Viruses commonly use programmed -1 ribosomal frameshifting to produce viral replicase. -1 frameshifting requires a slippery sequence just upstream of the in-frame stop codon and a structured RNA element several bases downstream of the slippery sequence. Ribosomal frameshifting has been demonstrated to be a means to increase the stability of PEMV2 RNA and its reporter transcript [1]. The frameshifting mechanism is also present in many important human viruses, such as SARS-CoV, HIV-1 and HIV-2, *Simian immunodeficiency virus*, and WNV [45], which are potential NMD targets. The underlying mechanism of ribosomal frameshifting is believed to remove viral RNA-bound UPF1 by the continuous movement of translating ribosomes.

### 3.3. Others

Alternative splicing is tightly controlled on HIV-1 genomic RNA when generating multiple mRNAs from single pre-mRNA species. Although HIV-1 viral RNA possesses NMD-inducing features, it can evade NMD by hijacking UPF1 [66,101]. UPF1 exerts multifaceted nature during HIV infection [102]. The viral protein pr55Gag is a structural protein that sharpens the nuclear transport of HIV-1 viral RNA. Interestingly, UPF1 co-localizes with the pr55Gag RNP complex, eventually intensifying viral replication [101]. Unexpectedly, UPF1 promotes HIV-1 viral RNA stability and translation [66], while UPF2 and SMG6 were found to inhibit HIV replication in CD4+ T cells [100] and primary monocyte-derived macrophages cells [102].

**Table 1 viruses-15-00816-t001:** Host NMD eliciting features and possible resistant factors of various viruses.

Virus Species	Types of Genome	Host	NMD-Eliciting Feature	NMD Resistant Factor	Reference
PVX	(+) ssRNA	*Arabidopsis thaliana*	iTC and long 3′ UTR	Unknown	[55]
TCV	(+) ssRNA	*Nicotiana benthamiana*	iTC and long 3′ UTR	Unstructured element	[55]
PEMV2	(+) ssRNA	*N. benthamiana*	Long 3′ UTR	p26 (MP), frameshifting element	[73]
Rotavirus	dsRNA	African green monkey kidney cell line	Unknown	non-structural protein 5 (NSP5)	[59]
CaMV	DNA	*N. benthamiana*	Polycistronic 35S pgRNA	Transactivator Protein (TAV)	[70]
HTLV-1	(+) ssRNA	Hela and HEK cell line	Unknown	TAX and REX factor	[21,64,65]
HCV	(+) ssRNA	hepatoma cells	Unknown	Core Protein	[82]
SFV	(+) ssRNA	Hela Cell line	Unknown	Replicase /nsp3 protein, capsid protein	[54,80,81,96]
MoMLV	(+) ssRNA	Mouse	Long 3′ UTR on *gag* gene	MoMLU-RT	[67]
CTFV	dsRNA	HEK293T cells	Unknown	Hairpin structure (CTFV-HP)	[99,100]
Zika Virus	(+) ssRNA	Human NPC cells	Unknown	Capsid Protein	[79]
RSV	(+) ssRNA	CEF cell line	uORF and long 3′ UTR	RNA Stability Element	[62]
HIV-1	(+) ssRNA	Hela cell line		REV	[101,103]
MHV	(+) ssRNA	17Cl-1 Cells, Mouse astrcytoma cell line	uORF and 3′ UTR	N Protein	[57]
KSHV	DNA	PEL cell line	Long 3′ UTR of *Orf50* mRNA	Unknown	[71,72]
EBV	DNA	HEK293T cells	Long 3′ UTR of BRLF1-encoding transcripts	Unknown	[71,72]

## 4. Concluding Remarks

The ample scenarios of the host NMD–virus interaction have been rooted out to a substantial measure. The diverse families of viruses evolve multifarious tactics to protect viral genomes from RNA decay and antiviral silencing during the infection cycle. Evolutionarily, to cope with host antiviral actions, (+) RNA viruses and retroviruses deploy *cis*- and *trans*-arm-based strategies to attenuate host antiviral immunity and facilitate viral replication [104]. Although the precise mechanism is not elucidated yet for most virus families, significant progress has been made in the molecular interaction of NMD and viruses in recent years.

Regarding DNA viruses that previously seemed unlikely to be associated with NMD, recent evidence shows that EBV and KSHV vigorously interact with host NMD components [71,72]. Long 3′ UTR and retained intron impose *Herpesvirus* RNAs to NMD. Whether DNA viruses encode anti-NMD factors or have any specific RNA sequence or structure that could resist NMD remains unsolved. On the other hand, questions can be raised that most (+) RNA viruses can potentially be an NMD target, but a significant number are still undiscovered. All (+) RNA viruses reported to be NMD targets have a monopartite viral genome that contains one or more iTC. Quite a portion of plant viruses is segmented in their genome, such as the family of *bromoviridea* that has a tripartite RNA genome. It would be interesting to investigate whether segmented viral genomes have the potential for NMD? Moreover, little is known about viruses whose genome is double-stranded RNA. Thus, numerous efforts will be required to shed light on taxonomically different viruses to create a whole picture of the NMD–virus molecular interaction.

NMD is a translation-dependent pathway. Alternative translation initiation factors such as VPg, 3′ cap-independent translation enhancer (CITE), and an internal ribosomal entry site (IRES) employed by viruses in various circumstances inside host cells may potentially impact the modulating or reshaping decay pathway, which have been overlooked till now. Viruses deploy virally induced cellular microenvironment to avoid the antiviral response [105]. What is the mechanism of action of NMD when a virus multiplies inside a virally encoded secured compartment in the cytoplasm or are there relations between NMD and viroplasms? Plenty of questions remain unanswered, which could expand our understanding of NMD and open new avenues for virus research in treating plant and animal diseases.

## Figures and Tables

**Figure 1 viruses-15-00816-f001:**
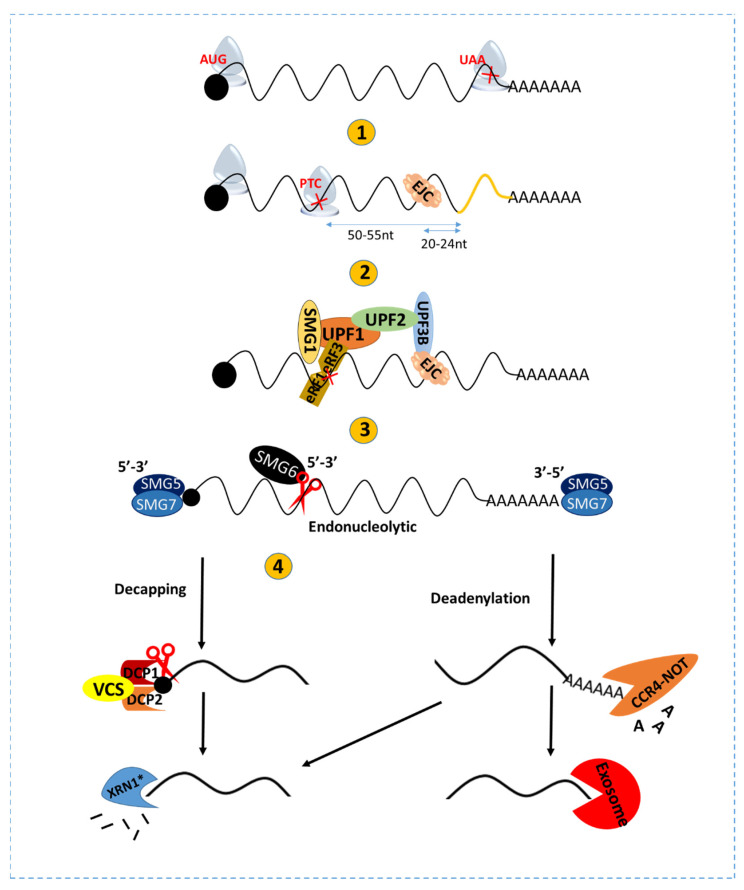
PTC-based NMD activation and RNA decay. When PTC is located >50-55 nucleotides upstream of the exon–exon junction. EJC deposits 20–24 nucleotides upstream of the exon–exon junction and cannot be removed by translating ribosomes. Subsequent deposition of eRF1 and eRF2 dissociates ribosome, thereby binding eRF3 and UPF1. SMG1-mediated phosphorylation of UPF1 initiates a conformational change of UPF1 to recruit UPF2 and UPF3, which precedes the reaction further. SMG1, UPF1, eRF1, and eRF3 together form the SURF complex. Afterward, UPF2 connects EJC and UPF1 via UPF3, which recruits relevant decay factors. Target transcripts can be degraded by two major cellular cleavage pathways, endo-nucleolytic and exo-nucleolytic. SMG6 is an endonuclease recruited by phosphorylated UPF1 at the PTC, cleaving the NMD substrate. SMG5/7 promote the exo-nucleolytic pathway by recruiting decapping complex DCP1a/DCP2 at the 5′ end and recruiting deadenylation complex CCR4-NOT at the 3′ end. Nascent decapped transcripts were further digested in a 5′→3′ manner via XRN1*. Deadenylated RNA transcripts can be degraded in two ways: exosome-mediated digestion in a 3′→5′ manner and XRN1*-mediated digestion in a 5′→3′ manner. [XRN1*: XRN1 functions in animal cells, while XRN4 does in plant cells].

**Figure 2 viruses-15-00816-f002:**
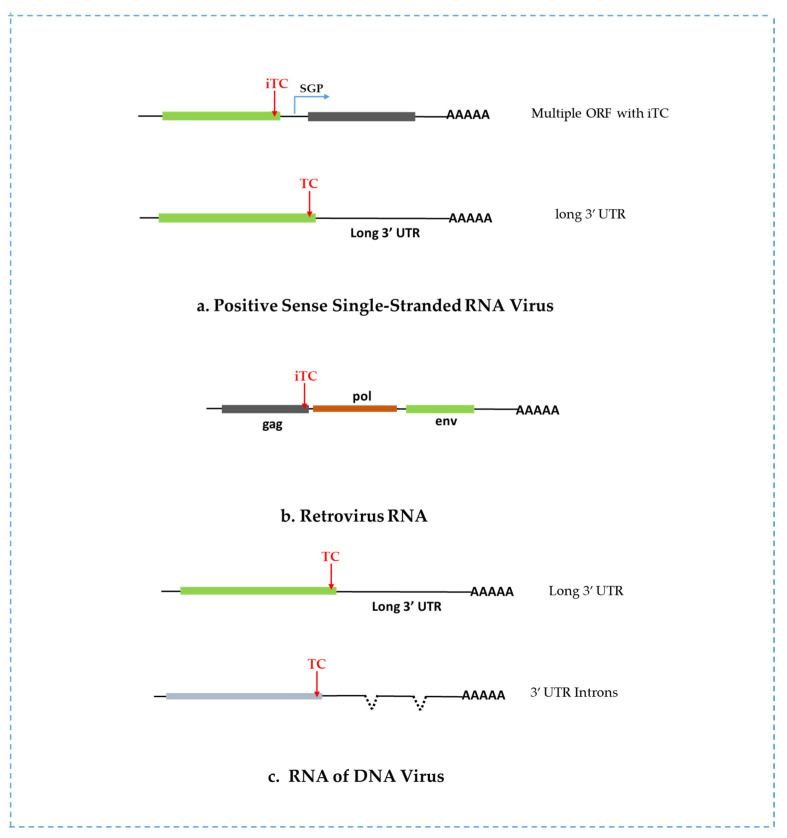
Schematic diagram of various viral RNAs targeted by host NMD. (**a**) Positive sense RNA virus possesses two or more separated open reading frames (e.g., *semliki forest virus* and *sindbis virus*) or a long 3′ UTR (e.g., *pea enation mosaic virus* 2); (**b**) Retroviral RNA (e.g., HIV, *Rous Sarcoma Virus*) harboring internal termination codons (iTC) in the *gag* gene; (**c**) Susceptible RNAs of DNA viruses (e.g., *Kaposi’s sarcoma-associated herpesvirus*, *Epstein-Barr virus*) with an extended 3′ UTR and introns [SGP: Sub-Genomic Promoter].

**Figure 3 viruses-15-00816-f003:**
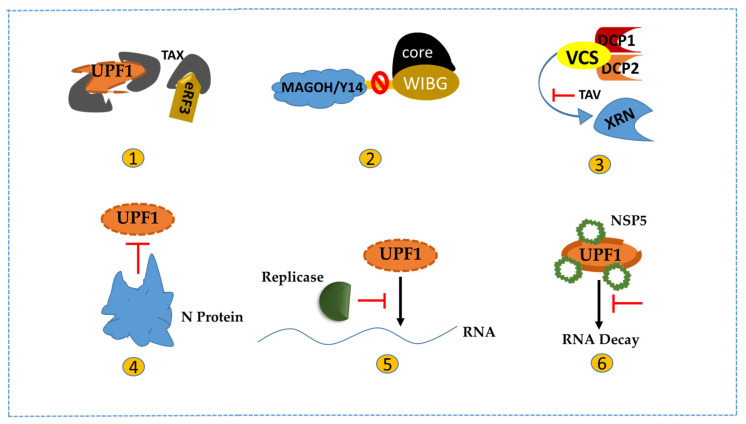
Viral protein-based strategies to modulate NMD activity: (**1**) Retroviral Tax factor antagonizes NMD by binding with eRF3 and UPF1 to prevent the SURF complex formation. By binding with UPF1, Tax disrupts its translocation and binding affinity to nucleic acids and other proteins; (**2**) HCV Core Protein disrupts NMD by binding with host protein WIBG to block the recycling interaction between WIBG and MAGOH/Y14; (**3**) Trans-acting activator (TAV) of CaMV interrupts decapping by interacting with VCS and subsequent decay of decapped transcripts by the XRN complex; (**4**) Coronavirus N-protein binds to UPF1 and diminishes its activity for NMD; (**5**) SFV replicase protein dislocates UPF1 from viral RNA to modulate host NMD response; (**6**) Non-structural protein (NSP5) promotes UPF1 degradation during *rotavirus* infection to inhibit the RNA decay pathway.

## Data Availability

Not applicable.

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
