# Peer review of "Molecular Interaction of Nonsense-Mediated mRNA Decay with Viruses"

_viruses, 2023, doi:10.3390/v15040816_

Round 1

Reviewer 1 Report

In the review, the author summarized the current NMD-mediated viral RNA degradation scenario and classified various molecular means by which viruses compromise the NMD-mediated antiviral defense for better infection in their hosts.  The review discussed the translation strategies of the viruses and described the viruses evade NMD medicated defense.  However, there are still some formats that should be corrected.

Line 199, line 318, and line 398, there are extra.   inside.

And the format of subtitles 3.1.1, 3.1.2, and 3.1.3 should be the same as the other parts.

Author Response

In the review, the author summarized the current NMD-mediated viral RNA degradation scenario and classified various molecular means by which viruses compromise the NMD-mediated antiviral defense for better infection in their hosts. The review discussed the translation strategies of the viruses and described the viruses evade NMD medicated defense. However, there are still some formats that should be corrected.

Reply: Thank you very much for your positive comment.  We have corrected all issues as shown below.    

1. Line 199, line 318, and line 398, there are extra.   inside.

Reply: Thank you for pointing out this issue. We have removed all extra dots (.) in the positions indicated, and also checked this issue in the whole manuscript.

2. And the format of subtitles 3.1.1, 3.1.2, and 3.1.3 should be the same as the other parts.

Reply: Thank you. Due to the major revision of the manuscript, we have removed 3.1.1, 3.1.2, 3.1.3 and reorganized into Section 2. All the subtitles in this section are formatted consistently.

Reviewer 2 Report

This review has summarized the current researches on nonsense-mediated mRNA decay (NMD), a cellular quality-control mechanism ensuring the translation fidelity, which is used by hosts to suppress virus infection. The summary on NMD mediated viral RNA degradation and the counter-defense of viruses on host NDM would help us to more fully appreciate the interaction between host and virus. However,the paper suffers for some serious limits:

1. The overall structure of the article needs to be adjusted to make it more logical. For example, the two parts 1. introduction and 2. Translation strategies of viruses need to be merged, and the relationship between virus translation strategies and NMD needs to be explained; Cauliflower Mosaic Virus (Pararetrovirus)(L185), as a kind of virus, is strange to be listed with 3.1 Positive Sense RNA Virus (L145), 3.3 Retroviruses and 3.4 Other (DNA viruses)(L218).

2. The title of the article needs to be revised to be easily understood. Why is RNA viruses emphasized in the title whereas DNA viruses are contained in the context?

3. Its hard to understand why various viruses are simply enumerated in the part of “3. NMD mediated defense against viruses” rather than to summarize the rule of NMD as a defense mechanism and the differences for various viruses.

4. More detailed information of Table 1 (L414) should be added, such as references, genome type of viruses and hosts. And its better to use three-line table.

5. The latest scientific research should be mentioned in this review, especially for the articles published in 2021-2022.

Author Response

This review has summarized the current research on nonsense-mediated mRNA decay (NMD), a cellular quality-control mechanism ensuring translation fidelity, which is used by hosts to suppress virus infection. The summary on NMD-mediated viral RNA degradation and the counter-defense of viruses on host NDM would help us to more fully appreciate the interaction between host and virus. However, the paper suffers from some serious limits:

Reply:   Thank you very much for your instructive comments and suggestions, which undoubtedly improved our manuscript.  Next are the responses to all of your comments one-by-one.

1.  The overall structure of the article needs to be adjusted to make it more logical. For example, the two parts “1. introduction” and “2. Translation strategies of viruses” need to be merged, and the relationship between virus translation strategies and NMD needs to be explained; Cauliflower Mosaic Virus (Pararetrovirus)(L185), as a kind of virus, is strange to be listed with “3.1 Positive Sense RNA Virus (L145)”, “3.3 Retroviruses” and “3.4 Other (DNA viruses)” (L218).

Reply: Thank you for your valuable advice. According to your comment, a major revision has been made to this manuscript. The “1. introduction” and “2. Translation strategies of viruses” have been merged. We have described the relationship between how translation strategies are important for NMD. For example, we have mentioned readthrough, frameshift, sub-genomic RNA and poly-protein strategies used by viruses to continue translation by compromising the NMD pathway. The detailed information is described in the last paragraph (lines 79-101) of the Introduction section. In addition, as a DNA virus, cauliflower mosaic virus (CaMV) is incorporated into the subsection of DNA viruses, which would make more sense to readers.

2.  The title of the article needs to be revised to be easily understood. Why are RNA viruses emphasized in the title whereas DNA viruses are contained in the context?

Reply:  Thank you for the important suggestion. Indeed,  this is our ignorance.  So, we have removed the word “RNA” from the title, which would make the manuscript more logical with the title.

3.  It’s hard to understand why various viruses are simply enumerated in the part of “3. NMD mediated defense against viruses” rather than to summarize the rule of NMD as a defense mechanism and the differences for various viruses.

Reply: Thank you for this comment. We have made a substantial change to this section. In the revised version (please see Section 2), we removed the subtitles of each virus and re-organized them based on their genome types, including (+) RNA viruses, dsRNA viruses, retroviruses, and DNA viruses. We hope this change makes this section reasonable and understandable.

4.  More detailed information in Table 1 (L414) should be added, such as references, genome type of viruses, and hosts. And it’s better to use a three-line table.

Reply: Thank you for this suggestion. All the information suggested has been added in Table 1, and also we made some essential corrections as highlighted. 

5.  The latest scientific research should be mentioned in this review, especially for the articles published in 2021-2022.

Reply: Thank you for pointing out the issue. We have added six relevant articles published in 2021-22, with reference numbers 21, 42, 59, 72, 82, and 105 in the revised version. 

Reviewer 3 Report

                This manuscript contains a dense overview of the interaction and interfacing of viral RNAs with the host NMD machinery.  Overall while the manuscript contains a great deal of information, it is also rather densely written that may make it difficult for the non-expert to fully digest and understand.  I have additional several specific suggestions to polish the presentation.

Major Points:

1.        All parts of the manuscript should be carefully reviewed for optimal usage of the English language. 

2.       The mixing of plant and animal viruses in the translation strategies section is confusing and incomplete.  Polyprotein strategies, negative sense RNA virus strategies, etc. are overlooked.  I would strongly recommend that the authors split this section up into animal virus translation strategies and plant virus strategies – particularly since clearly establishing these strategies for the reader is vital for understanding of the potential of the RNAs as NMD substrates.

3.       Alphavirus NMD defenses:  The manuscript is missing a key recent paper (Contu et al PloS Pathogens 2021) that demonstrates the viral capsid protein inhibits NMD

Minor Points:

1.        Abstract:  Is it really true that the majority of viral RNAs are polycistronic?  ORFs that encode polyproteins are arguably technically polycistronic since there is no intercistronic region.

2.       Introduction:  Since PTCs still trigger translation termination, referring to them as ‘Failures in successful translatin termination…’ is not accurate.

3.       Fig.1:  The depiction of a 5’-3’ exonuclease acting on the cap (drawn in a way that is different from the decapping complex) is misleading and may be very confusing to readers.  Thus Step 3 needs to be redrawn.  In addition, Xrn1 should be depicted acting on both fragments shown at the bottom of figure (as it is responsible for degrading endonucleolytic fragments in a 5’-3’ direction).

4.       Heading 3.3 and 4.3 have an extra period

5.       Line 280:  West Nile virus is misspelled

6.       Line 337:  It is not clear why reference 102 (a retrovirus paper) is included in this section on SFV

Author Response

This manuscript contains a dense overview of the interaction and interfacing of viral RNAs with the host NMD machinery. Overall while the manuscript contains a great deal of information, it is also rather densely written that may make it difficult for the non-expert to fully digest and understand. I have additional several specific suggestions to polish the presentation.

Reply:   Thank you very much for your instructive comments and suggestions, which undoubtedly improved our manuscript.  Next is the responses to all of your comments one-by-one.

Major Points:

1.  All parts of the manuscript should be carefully reviewed for optimal usage of the English language.

Reply:  Thank you so much for your precious comment. According to your comment, we have revised the English language carefully in the entire manuscript. We also used the software “Grammarly” to correct the grammatical errors. We hope the revised version is good in English for publication.

2.  The mixing of plant and animal viruses in the translation strategies section is confusing and incomplete. Polyprotein strategies, negative-sense RNA virus strategies, etc. are overlooked. I would strongly recommend that the authors split this section up into animal virus translation strategies and plant virus strategies – particularly since clearly establishing these strategies for the reader is vital for an understanding of the potential of the RNAs as NMD substrates.

Reply: We have made comprehensive revisions to this section as both reviewers suggested to improve this part. We have merged the “translation strategies” with the “introduction” section and described how translation strategies are significant for NMD pathway execution. As the negative-sense RNA viruses are not involved in NMD until now, we omitted the translation strategies of negative-sense RNA viruses from the manuscript. The strategy of producing polyprotein precursor strategy is used commonly by human, animal, and plant viruses.  Whatever, in the revised version, the text about the polyprotein strategy has been compromised as “Some viral genomes (e.g., polioviruses, HIV, Dengue virus, HCV, potyviruses) encode a large ORF that is translated to produce a polyprotein precursor. The precursor is cleaved by cellular or viral protease enzymes to produce all functional viral proteins [46]. Such viruses use the strategy of producing a polyprotein precursor to avoid the presence of any internal stop codon (iTC) in the large ORF, which would protect viral RNAs from NMD.”.

3.  Alphavirus NMD defenses: The manuscript is missing a key recent paper (Contu et al PloS Pathogens 2021) that demonstrates the viral capsid protein inhibits NMD.

Reply: We are grateful to you for providing the recently published paper. We have added the paper in Section 3.1.3  (line numbers 274-277) with reference No. 82 in the manuscript.

Minor Points:

1.  Abstract: Is it really true that the majority of viral RNAs are polycistronic? ORFs that encode polyproteins are arguably technically polycistronic since there is no intercistronic region.

Reply: Thanks for raising the question.  We think the wording is not accurate for the majority.  Here we change “the majority of”  to “many”.  We agree that it is easily confused with polycistronic and polyprotein.  Polycistroic is usually used to describe prokaryotic RNA transcripts that encode two or more coding sequences.  Since viral genome is highly compacted, many RNA viral genomes encode multiple open reading frames.  Polyproteins we mentioned in the manuscript refer to the polyprotein precursor that is translated from one ORF and is not essentially associated with polycistronic RNAs.  To avoid confusion, we made some changes in the revised version. For an instance, “a majority of RNA viruses are polycistronic” is replaced with “Many RNA viruses have a genome that contains internal stop codon(s)” in the Abstract.

2.  Introduction: Since PTCs still trigger translation termination, referring to them as ‘Failures in successful translatin termination…’ is not accurate.

Reply:  Thank you for mentioning this issue. We have replaced the ‘Failures in successful translation termination…’ with “Premature termination accelerates the destruction of concerned mRNA through the NMD pathway” (line number 41-42).

3.  Fig.1: The depiction of a 5’-3’ exonuclease acting on the cap (drawn in a way that is different from the decapping complex) is misleading and may be very confusing to readers. Thus Step 3 needs to be redrawn. In addition, Xrn1 should be depicted acting on both fragments shown at the bottom of figure (as it is responsible for degrading endonucleolytic fragments in a 5’-3’ direction).

Reply:  Thanks again. We have redrawn Figure 1 according to your suggestions. We have shown that SMG5/7 acts on the 3’ and 5’ end of NMD targeted transcripts and initiates the degradation activity by recruiting decay factors in both ends. The depiction of XRN1 is updated in the revised version of the manuscript according to the comment. Moreover, we also show the deadenylated transcript is further degraded by the exosome in a 3’-5’ manner.

4.  Heading 3.3 and 4.3 have an extra period.

Reply: it has been removed.

5.  Line 280: West Nile virus is misspelled

Reply: It has been corrected. We have corrected the spelling of West Nile Virus in the updated manuscript (now line number 260).

6.  Line 337: It is not clear why reference 102 (a retrovirus paper) is included in this section on SFV

Reply: Thank you so much for figuring out this error. We have removed this reference.

Round 2

Reviewer 2 Report

I have checked up the changes of the manuscript(viruses-2128463)entitled "Molecular Interaction of Nonsense Mediated mRNA Decay with RNA Viruses. The authors have made appropriate responses to the review comments in their revised manuscript, and I feel that they have satisfied the previously documented concerns. I am pleased to recommend the manuscript for publication in Viruses.

Reviewer 3 Report

                The revised manuscript is improved and addresses the points raised in the original round of critiques.